

# Region-of-interest analyses of one-dimensional biomechanical trajectories: bridging 0D and 1D theory, augmenting statistical power

Todd C. Pataky[1], Mark A. Robinson[2] and Jos Vanrenterghem[3]

[1] Institute for Fiber Engineering, Department of Bioengineering, Shinshu University, Ueda, Nagano, Japan
[2] Research Institute for Sport and Exercise Sciences, Liverpool John Moores University, Liverpool, United Kingdom
[3] Department of Rehabilitation Sciences, Katholieke Universiteit Leuven, Belgium

## ABSTRACT

One-dimensional (1D) kinematic, force, and EMG trajectories are often analyzed using zero-dimensional (0D) metrics like local extrema. Recently whole-trajectory 1D methods have emerged in the literature as alternatives. Since 0D and 1D methods can yield qualitatively different results, the two approaches may appear to be theoretically distinct. The purposes of this paper were (a) to clarify that 0D and 1D approaches are actually just special cases of a more general region-of-interest (ROI) analysis framework, and (b) to demonstrate how ROIs can augment statistical power. We first simulated millions of smooth, random 1D datasets to validate theoretical predictions of the 0D, 1D and ROI approaches and to emphasize how ROIs provide a continuous bridge between 0D and 1D results. We then analyzed a variety of public datasets to demonstrate potential effects of ROIs on biomechanical conclusions. Results showed, first, that *a priori* ROI particulars can qualitatively affect the biomechanical conclusions that emerge from analyses and, second, that ROIs derived from exploratory/pilot analyses can detect smaller biomechanical effects than are detectable using full 1D methods. We recommend regarding ROIs, like data filtering particulars and Type I error rate, as parameters which can affect hypothesis testing results, and thus as sensitivity analysis tools to ensure arbitrary decisions do not influence scientific interpretations. Last, we describe open-source Python and MATLAB implementations of 1D ROI analysis for arbitrary experimental designs ranging from one-sample $t$ tests to MANOVA.

## INTRODUCTION

Many biomechanical measurements may be regarded as '$n$-dimensional $m$-dimensional' ($nDmD$) continua, where $n$ and $m$ are the dimensionalities of the measurement domain and dependent variable, respectively. Common examples include: joint flexion (1D1D), ground reaction force (1D3D), plantar pressure distribution (2D1D) and bone strain tensor distributions (3D6D). These data are often analyzed using 0D1D metrics from regions of

Corresponding author
Todd C. Pataky,
tpataky@shinshu-u.ac.jp

interest (ROIs) which summarize particular continuum features. In this paper 'ROI' refers to a geometrical subset of a continuum dataset, and 'ROI analysis' refers to the analysis of data extracted from an ROI. More explicit definitions for these terms with literature context are provided in Appendix A.

In $n > 1$ datasets ROIs are often explicitly constructed based on anatomical rationale, especially for plantar pressure (*Cavanagh & Ulbrecht, 1994*) and finite element analyses (*Radcliffe & Taylor, 2007*). In $n = 1$ datasets ROIs tend to be used both explicitly (e.g., with phase labels including: "early stance," "push off," "swing," etc.) (*Blanc et al., 1999*) and implicitly (e.g., local extrema are used without explicitly labeled continuum regions) (*Cavanagh & Lafortune, 1980*). Regardless, the ultimately analyzed metrics are often $n = 0$ scalars, so we refer to this class of methods as '0D.' For simplicity the remainder of this paper focusses on $n = 1$ datasets and corresponding '1D methods' (Appendix A).

Recently a variety of 1D methodologies have emerged in the Biomechanics literature including functional data analysis (FDA) (*Ramsay & Silverman, 2005*), principal component analysis(PCA) (*Daffertshofer et al., 2004*) and statistical parametric mapping (SPM) (*Pataky, Robinson & Vanrenterghem, 2013*), each of which afford whole-field 1D$m$D analysis. SPM in particular is ideal for ROI-related hypothesis testing because it is valid for arbitrary 1D geometries including broken or segmented regions of arbitrary size (*Pataky, 2016*). FDA is less ideal because it employs continuous basis functions and not, to our knowledge, piecewise continuous ones. PCA can easily handle arbitrary ROI data, but it is predominantly a data reduction technique and not a hypothesis testing technique.

This paper therefore focusses on SPM, a methodology that was initially developed in the Neuroimaging literature in the 1990s (*Friston et al., 1995*), that spread to Electrophysiology through the 2000s (*Kiebel & Friston, 2004*; *Kilner, Kiebel & Friston, 2005*), and which has more recently appeared in the Biomechanics literature (*Pataky, 2012*; *Pataky, Robinson & Vanrenterghem, 2013*). In Neuroimaging and Electrophysiology SPM has grown into a comprehensive suite of techniques capable of handling all aspects of $n$-dimensional continuum analysis including univariate and multivariate continuum analysis, parametric and non-parametric probability density utilization, classical and Bayesian inference, and multi-modal analysis among other functionality (*Friston et al., 2007*). In the context of this paper, SPM's classical hypothesis testing ability is key. Briefly, and considering only 1D data, SPM first computes a 1D test statistic continuum (often the $t$ statistic continuum) from a set of experimentally measured 1D continua. This step is effectively equivalent to 1D mean and standard deviation continuum computation, which is frequently done in the Biomechanics literature. SPM then conducts statistical inference at a Type I error rate of $\alpha$ by calculating the critical test statistic value above which random test statistic continua (generated by smooth, 1D Gaussian continua) would traverse in only $(1 - \alpha)\%$ of an infinite number of identical experiments; if the experimentally observed continuum exceeds that critical value the null hypothesis is rejected. This general approach to classical hypothesis testing has been validated extensively in the Neuroimaging literature for 3D (and 4D) continua (*Friston et al., 1995*; *Friston et al., 2007*) and has also been validated for 1D univariate and multivariate data (*Pataky, 2016*).

Despite the validity of 1D approaches, a variety of conceptual difficulties may arise when attempting to corroborate 0D and 1D approaches. In particular, 0D statistical results are typically tabulated using single numbers for the test statistic and $p$ values, so test statistic continua may appear odd. Another apparent discrepancy between 0D and 1D techniques is that the former requires continuum summary metric extraction but the latter does not, so 0D techniques may appear to be somewhat more subjective than 1D techniques. A final discrepancy is that 1D techniques involve multiple comparison corrections, so they may appear to be less powerful than 0D techniques. All of these real or perceived discrepancies could lead one to infer that 0D and 1D approaches are fundamentally different.

The primary purpose of this paper was to clarify the theoretical consistency between 0D and 1D techniques as special cases of ROI analysis. To that end we describe 1D ROI theory then validate its predictions using numerical simulations of random datasets with temporal scopes ranging from single points to large 1D continua. The second purpose was to demonstrate how ROIs can be used to augment statistical power in both exploratory and hypothesis-driven experiments. The final purpose was to introduce an open-source software implementation of ROI analysis (in Appendix C) which emphasizes how 0D, 1D and ROI analyses can all be executed using a common software interface.

## METHODS

All analyses were implemented in Python 2.7 (Van Rossum, 2014) using Canopy 1.6 (Enthought Inc., Austin, TX, USA) and spm1d (Pataky, 2012; Pataky, 2016) (http://www.spm1d.org). All datasets described below are included in the **spm1d** package and are accessible using the **spm1d.data** interface as described in Appendix D. High-level Python and MATLAB (The MathWorks, Natick, MA, USA) interfaces for ROI analyses are now available in spm1d as described in Appendix C.

### ROI theory and validation

In classical hypothesis testing the null hypothesis is rejected if the experimentally observed test statistic $t$ exceeds a critical threshold $t^*$, which can be computed according to:

$$P(t > t^*) = \alpha \tag{1}$$

where $\alpha$ is the Type I error rate (usually 0.05) and $P(t > t^*)$ is the probability that the test statistic exceeds $t^*$ if the null hypothesis is true.

For an ROI of size $S$ ($S \geq 0$), Eq. (1) can be written as (Friston et al., 2007; Pataky, 2016):

$$P(t_{\max} > t^*) = 1 - \exp\left[-P_{0D}(t > t^*) - \frac{S}{W} \frac{\sqrt{4\log 2}}{2\pi} \left(1 + \frac{(t^*)^2}{\nu}\right)^{-(\nu-1)/2}\right] = \alpha \tag{2}$$

where $t_{\max}$ is the maximum value of the $t$ statistic inside the ROI, $P_{0D}(t > t^*)$ is the probability under the null hypothesis that 0D random Gaussian data will produce a $t$ value greater than $t^*$, $W$ is the FWHM representing trajectory smoothness (Appendix B), and $\nu$ is the degrees of freedom. Note that $P(t_{\max} > t^*)$ converges to $P_{0D}(t > t^*)$ as $S$ approaches zero, and that $t^*$ must increase as $S$ increases to maintain a given $\alpha$. In other words, the larger the

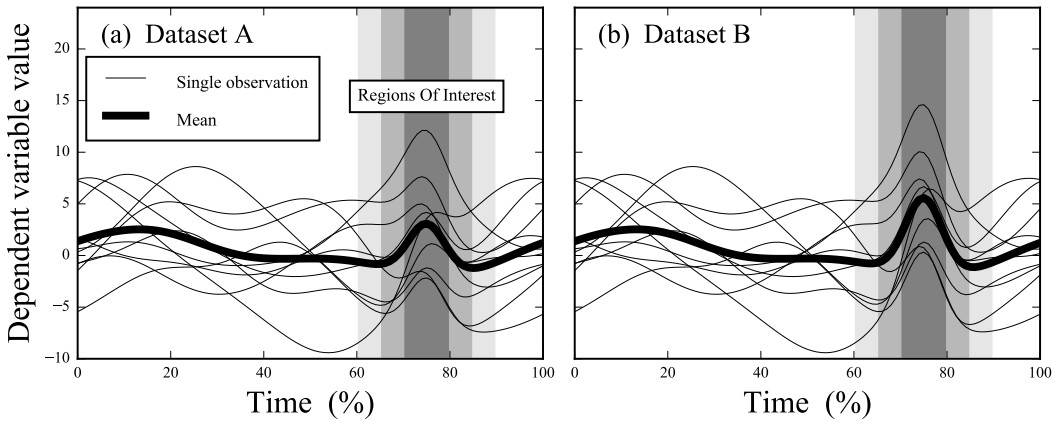

**Figure 1** **Simulated datasets.** Datasets A and B are identical except in Dataset B the signal at time = 75% is amplified. Three regions of interest (ROIs) are depicted, centered at time = 75% and spanning time windows of 10%, 20% and 30%, respectively.

ROI, the more likely smooth, purely random 1D Gaussian data are to produce high $t$ values. Also note that Eq. (2) can accommodate multiple ROIs using a relatively simple correction (*Friston et al., 2007*).

We computed $t^*$ for a range of ROI sizes ($S$), smoothness values (FWHM) and sample sizes ($v$). ROI size was systematically varied over the full range of possibilities (i.e., from 0 to 100% trajectory length). FWHM and $v$ values were selected to span a range of values observed in representative open-access biomechanical datasets (FWHM = [5, 50], $v = [5, 49]$) (*Pataky, Vanrenterghem & Robinson, 2016*).

To validate Eq. (2) for arbitrary ROI sizes we simulated 100,000 smooth, purely random Gaussian 1D datasets using 'rft1d' (*Pataky, 2016*), and repeated for each combination of $S$, FWHM and $v$ values. For each random dataset we computed $t_{\max}$, thereby producing one distribution of 100,000 $t_{\max}$ values for each combination of parameters. We then estimated $t^*$ for each distribution as the 95th percentile of the distribution, then qualitatively compared to the theoretical result (Eq. (2)).

## Example ROI analyses
### Simulated datasets
Datasets A and B (Fig. 1) (*Pataky, Robinson & Vanrenterghem, 2013*) consisted of ten simulated, smooth, random 1D Gaussian fields to which a Gaussian pulse was added at time = 75%. The degrees of freedom and number of time nodes were $v = 9$ and $Q = 101$, respectively, for both datasets. The pulse was slightly larger in Dataset B than in Dataset A. Both were analyzed using six procedures, in order of increasing conservativeness: (1) 0D analyses on the local maxima at time = 75%, (2–4) 1D ROI analysis with ROIs centered at time = 75% and with temporal sizes of ±5%, ±10% and ±15%, respectively, (5) 1D full-field analysis (i.e., with ROI size = 100%), and (6) 1D full-field analysis with a Bonferroni correction. The latter assumes independence amongst adjacent trajectory nodes so is overly conservative for smooth data (*Pataky, Robinson & Vanrenterghem, 2013*).

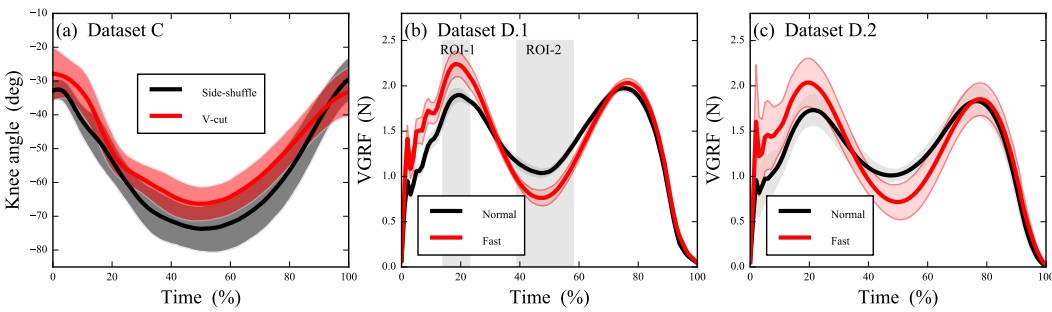

**Figure 2** **Experimental datasets; error clouds depict standard deviations.** (A) Knee kinematics during side-shuffle and v-cut maneuvers (*Neptune, Wright & Van den Bogert, 1999*). (B) Pilot data from $N = 1$ subject comparing vertical ground reaction force (VGRF) in "Normal" vs. "Fast" walking (*Pataky et al., 2008*). (C) Subsequent identical VGRF experiment conducted on $N = 6$ subjects (*Pataky et al., 2008*).

### *Experimental datasets*

Dataset C (Fig. 2A) (*Neptune, Wright & Van den Bogert, 1999*) ($\nu = 7, Q = 101$) contained stance-phase sagittal plane knee angles from eight participants who each performed both side-shuffle and v-cut maneuvers. We started with 0D analysis of maximal knee flexion (i.e., $S = 0$), then conducted three ROI analyses with ROIs centered approximately on maximal flexion (time = 50%) and with temporal extents of 10%, 40% and 80%, respectively. Finally, the 1D full-field ROI and Bonferroni procedures were applied as in the simulated datasets.

Dataset D (Figs. 2B and 2C) (*Pataky, Robinson & Vanrenterghem, 2013*) ($Q = 101$) contained stance-phase body-weight-normalized vertical ground reaction forces (VGRF) from seven subjects who each performed normal, self-paced walking and fast walking in a randomized order. Analysis proceeded in two-stages to demonstrate how ROIs can be used to increase analysis sensitivity in exploratory experiments (*Pataky, Vanrenterghem & Robinson, 2016*, Fig.7). First, the chronologically first subject was separated as a pilot subject (Fig. 2B). This subject's results were examined qualitatively and were used to define ROIs. Finally, those ROIs were used as an *a priori* constraint in analysis of the six remaining subjects (Fig. 2C) ($\nu = 5$). Results of this ROI-driven two-stage procedure were compared to a full-field 1D analysis.

## RESULTS

Critical thresholds $t^*$ necessary to maintain $\alpha = 0.05$ (Eq. (2)) increased nonlinearly as ROI size increased and $t^*$ values for 1D data converged to 0D $t^*$ values as ROI size approached zero (Fig. 3). Numerical simulations validated theoretical $t^*$ values for arbitrary 1D field smoothness values and arbitrary sample sizes. These results emphasize that 0D analyses are a special case of 1D analysis for which ROI size is zero.

In Dataset A, the test statistic exceeded the critical threshold for 0D analysis and also for a narrow ROI of 10%, but failed to reach the thresholds for wider ROIs of 20% and 30%, and also failed to reach the full-field threshold (Fig. 4A). In contrast, Dataset B's slightly amplified signal at time = 75% exceeded all thresholds except for the highly conservative

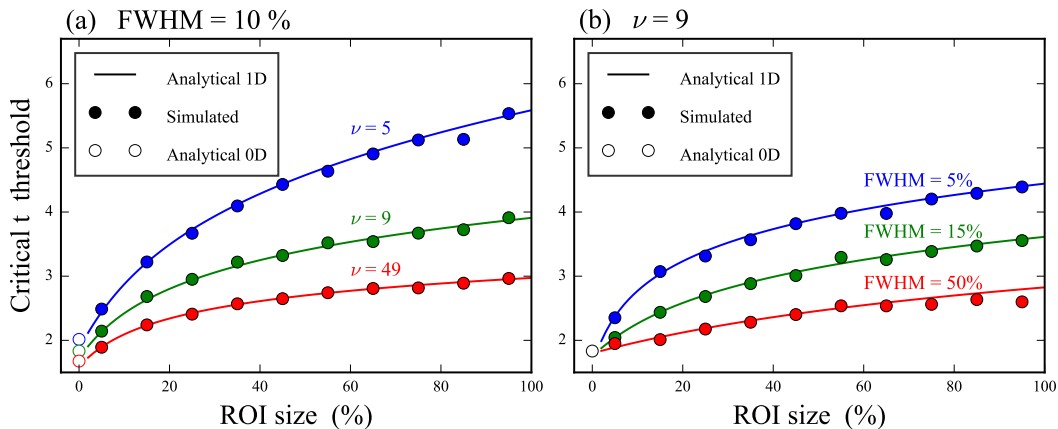

**Figure 3** **Validation of theoretical critical thresholds for region of interest (ROI) analysis.** (A) Three different degrees of freedom ($\nu$) and one smoothness (FWHM) value. (B) Three different FWHM values and one $\nu$ value. The analytical 0D results assume 0D Gaussian randomness and the 1D results assume smooth 1D Gaussian randomness.

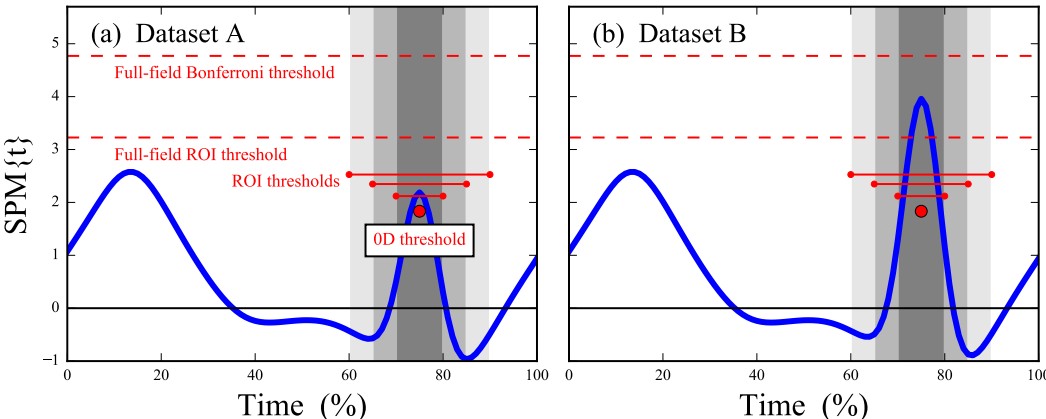

**Figure 4** **Simulated dataset hypothesis testing results.** "SPM{$t$}" denotes the $t$ value extended in time to form a 'statistical parametric map.' Six critical thresholds are depicted (see text). The width of each threshold depicts the temporal extent of the null hypothesis.

full-field Bonferroni correction (Fig. 4B). These results imply that small ROIs can identify relatively small effects like in Dataset A, and that large or even full-field ROIs can identify large effects. ROIs thus embody a trade-off between statistical power and the temporal scope of the null hypothesis; statistical power decreases as ROI size increases and vice versa.

For Dataset C, 0D analysis conducted on maximum knee flexion passed the critical threshold, as did a moderately broad ROI of 40% (Fig. 5). However, an ROI of 80% and full-field analysis failed to cross the critical threshold in the vicinity of maximum flexion. The null hypothesis was nevertheless rejected for both an ROI size of 80% and full-field analysis, but in this case the statistical conclusion pertains to regions other than maximum flexion.

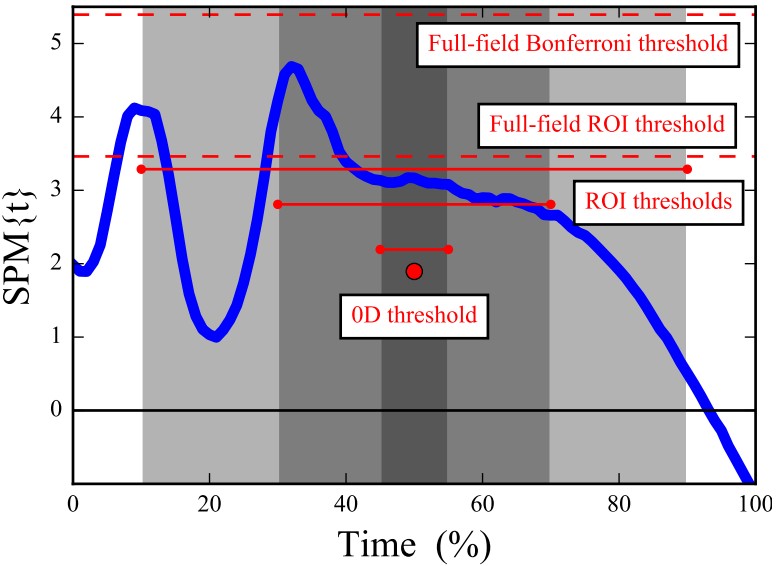

**Figure 5  Dataset C (knee flexion/extension) hypothesis testing results.** "SPM{*t*}" denotes the *t* value. Critical thresholds are presented as in Fig. 4, with both the height and temporal extent of each threshold depicted.

For Dataset D the pilot subject's data were used to identify two relatively narrow ROIs in the vicinity of the first local maximum and the local minimum at mid-stance (Fig. 6A). These ROIs led to null hypothesis rejection for both ROIs in the independent six-subject dataset (Fig. 6C). Had the ROIs not been defined the null hypothesis would not have been rejected (Fig. 6B).

## DISCUSSION

The main purpose of this paper was to clarify the theoretical consistency between "0D" and "1D" analyses, and to emphasize that both are actually just special cases of ROI analysis. In particular, the ROI size ($S$) and 1D smoothness (FWHM) parameters together construct a continuous theoretical bridge between 0D and 1D methodologies (Eq. (2), Fig. 3). ROI-based statistical analysis of $n$D continua was formalized in the Neuroimaging literature (*Brett et al., 2002*; *Poldrack, 2006*; *Friston et al., 2007*) where established ROI software tools exist, including especially MarsBar (*Hammers et al., 2002*; *Brett et al., 2002*) for SPM99 (Wellcome Trust Centre for Neuroimaging, University College London, UK).

Although ROI analyses are common in biomechanical applications like plantar pressure analysis (*Cavanagh & Ulbrecht, 1994*) and finite element analysis (*Radcliffe & Taylor, 2007*), statistical ROI theory has not, to our knowledge, been addressed in the Biomechanics literature.

The key theoretical point to consider when implementing ROI analyses is that small ROIs can more readily detect true within-region effects than large ROIs (Figs. 3 and 4). However, ROI analysis are not necessarily more sensitive than full-field 1D analysis because effects may exist outside the ROI (Fig. 5, narrowest ROI). An investigator must therefore balance local signal detectability (via narrow *a priori* ROI definition) with full-field signal

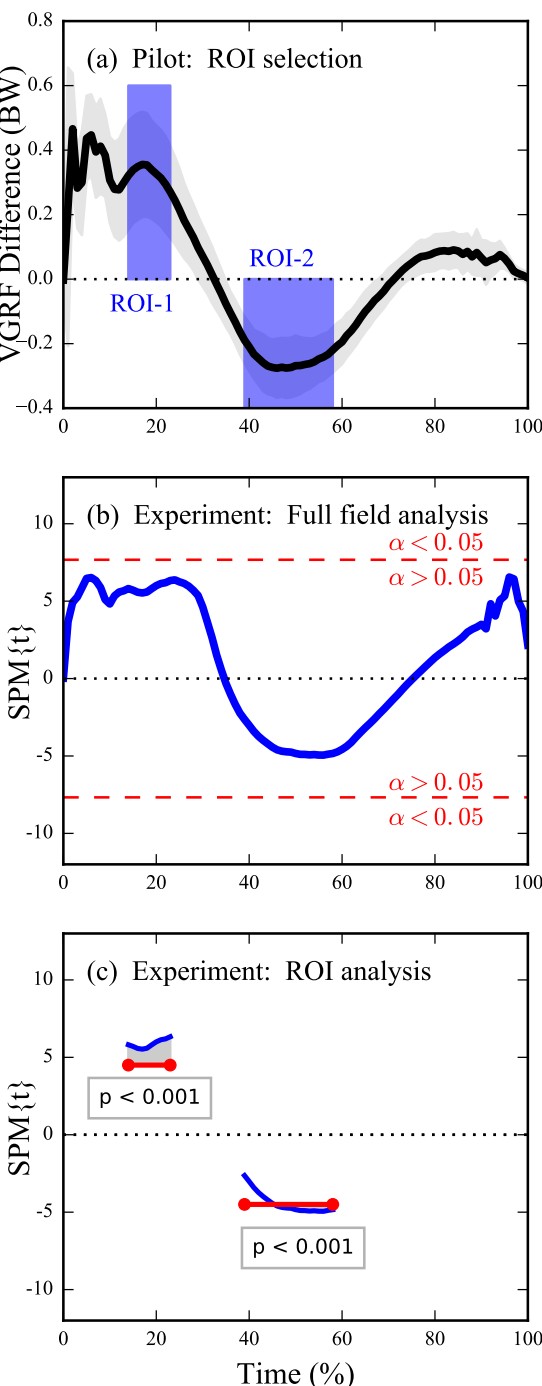

**Figure 6  Dataset D (VGRF) sensitivity augmentation procedure and results.** (A) Pilot study's mean inter-condition VGRF difference (see Fig. 2B) for $N = 1$ subject along with user-defined ROIs. The ROIs specify both the direction and temporal extent of an omnibus a priori hypothesis to be tested in an independent experiment. (B) Experimental analysis on $N = 6$ independent subjects using full-field inference (i.e., had no ROIs been selected); "SPM{$t$}" denotes the $t$ value. The experimental differences between Normal and Fast are insufficiently large to reject the full-field null hypothesis (because the $t$ value fails to traverse the critical threshold—depicted in red—at $\alpha = 0.05$). (C) Experimental analysis using ROI-based inference: the omnibus ROI-based null hypothesis is rejected.

detectability (via larger ROIs). In the absence of an *a priori* hypothesis regarding a specific portion of the continuum it has been suggested that full-field analyses should be conducted (*Pataky, Vanrenterghem & Robinson, 2016*), and this paper extends that suggestion to include a caveat for exploratory work in a two-stage procedure involving an initial full-field analysis on an exploratory dataset followed by more precise ROI-based testing (Fig. 6).

An apparent limitation of ROIs is that, since they can create / eradicate statistical significance (Figs. 4 and 5), they have the potential to be abused. We don't regard this limitation as unique to ROI analysis because ROI size, like $\alpha$, can be manipulated to artificially raise / lower critical thresholds. We would therefore recommend that, instead of choosing a single ROI size or a single $\alpha$, investigators should actively manipulate all parameters that might affect ultimate conclusions including: ROI size, $\alpha$, data filtering, coordinate system definitions, etc., and then actively report the results of those manipulations in a sensitivity analysis as has been done elsewhere (*Pataky et al., 2014*). If the reported results are robust to those manipulations then one can be more confident that the reported results are neither false positives (*Pataky, Vanrenterghem & Robinson, 2016*) nor false negatives. Such manipulations may be especially important for biomechanical datasets considering that these data can be sensitive to ROI definitions (Figs. 5–6), and also considering that some controversies regarding the relative merits of 0D and ROI vs. full-field analysis exist in other literatures (*Kubicki et al., 2002*; *Giuliani et al., 2005*; *Furutani et al., 2005*; *Friston et al., 2006*; *Saxe, Brett & Kanwisher, 2006*; *Snook, Plewes & Beaulieu, 2007*; *Kilner, 2013*).

More formally, ROI definitions and procedures should be considered from the perspective of 'circular analysis' (*Kriegeskorte et al., 2009*), an umbrella term encompassing bias-generating factors in scientific intepretations of processed data. In particular, the approach we have recommended with Dataset D — ROI generation based on independent-subject pilot studies—is analogous to 'functional localizers' in the Neuroimaging literature for which substantial benefits but also substantial cicular risks exist (*Friston et al., 2006*; *Saxe, Brett & Kanwisher, 2006*; *Kilner, 2013*). As a simple example of circular analysis and its dangers consider the local maximum in Dataset A near time = 15% (Fig. 4A). Defining an ROI about this maximum after seeing this result would lead to null hypothesis rejection even for relatively broad ROIs of 20%. This would be a circular conclusion because the observed result has directly affected the conclusion's assumption that the identified ROI was not of *a priori* interest but is now. In contrast, the 1D (full-field) threshold correctly fails to reject the null hypothesis; the 1D approach's conclusion is not circular because the *a priori* assumption of full-field effects was not affected by the observed result.

The literature contains a variety of recommendations regarding avoiding bias associated with circularity in ROI analyses, and recent developments in particular show that it is possible to use algorithmic data-driven ROI selection in an unbiased manner to increase statistical power and also maintain control over Type I error rates (*Brooks, Zoumpoulaki & Bowman, in press*). This result is limited to specific cases of experimental variance, but may nevertheless be promising for researchers concerned regarding inadequate power in analyses of large 1D datasets with potentially many 1D variables. Regardless, if one wishes to conduct ROI analysis or otherwise reduce the recorded 1D dataset, they should be aware

of circularity and how circular (il)logic can produce a vast array of biased conclusions following dataset reduction.

A technical point not addressed in the analyses above is how to handle within-ROI signals that extend beyond the ROI. This situation is observable in Fig. 6C, when the test statistic value is greater than the threshold on the ROI boundaries. From a classical hypothesis perspective the point is moot because the null hypothesis is rejected regardless of the temporal scope of a supra-threshold signal. It has been recommended elsewhere that, if a within-ROI signal is detected, the entire signal be reported even if it extends outside of the ROI (*Friston et al., 2007*). Our recommended approach of manipulating ROI location and size in a sensitivity analysis is consistent with that approach in that ROI boundaries should generally be regarded as soft.

In summary, this paper has introduced and validated an ROI approach for analyzing 1D biomechanical trajectories which clarifies the consistency between common 0D approaches and recent 1D approaches. Since biomechanical interpretations can be sensitive not only to ROI size but also to other data processing particulars like filtering and coordinate system definitions, it is recommended that ROIs be used only when there is adequate *a priori* justification for ignoring other regions of the 1D continuum, and that when they are used ROI sensitivity results are also reported.

### Funding

This work was supported by Wakate A Grant 15H05360 from the Japan Society for the Promotion of Science. The funders had no role in study design, data collection and analysis, decision to publish, or preparation of the manuscript.

### Grant Disclosures

The following grant information was disclosed by the authors:
Japan Society for the Promotion of Science: 15H05360.

### Competing Interests

The authors declare there are no competing interests.

### Author Contributions

- Todd C. Pataky conceived and designed the experiments, analyzed the data, wrote the paper, prepared figures and/or tables, reviewed drafts of the paper.
- Mark A. Robinson and Jos Vanrenterghem conceived and designed the experiments, analyzed the data, wrote the paper, reviewed drafts of the paper.

### Data Availability

All raw data analyzed in this paper are available in the "spm1d" software package available at: http://www.spm1d.org.

## Supplemental Information

Supplemental information for this article can be found online at http://dx.doi.org/10.7717/peerj.2652#supplemental-information.

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
