# Peer review of "Region-of-interest analyses of one-dimensional biomechanical trajectories: bridging 0D and 1D theory, augmenting statistical power"

_PeerJ, doi:10.7717/peerj.2652_

## Round 0.1 · original submission · Major Revisions

We are very sorry how long it took to get reviews; it has been an unusually difficult paper and time period to get reviewers for. But at last we have these two constructive reviews, and they constitute moderate revisions, which may or may not need re-review. Please include a detailed Response document with your resubmission, which should address every point of the reviewers individually. Thank you!

Reviewer 1 ·

Basic reporting

Overall, the manuscript is well concise, well-written, and does a good job in explaining and answering the research questions. The submission is self-contained and clear in its purpose and meeting this purpose.

For the most part, all aspects of the manuscript are well explained and supported by a sound rationale or experimental data. Given that the paper focuses on the SPM method of 1D analysis, it may be useful to include some additional info / references to appropriate papers for this method in the introduction. There is a general discussion on 1D methods within the introduction of the manuscript, however given the focus on SPM some more details on this 1D method may be useful for those unfamiliar with the method. A brief description of the method, similar to Pataky (2012), or references to earlier work published by this research group to direct readers to this information would likely be sufficient.

A part from this point in the introduction, the reporting of data and findings throughout the manuscript is exceptional. The combination of in-text explanations with appropriate use of figures throughout makes some potentially difficult analytical concepts and findings easy to understand. For example, Figure 3 provides a nice summary of the relationship between increasing 1D ROI size and critical statistical thresholds / statistical power relative to 0D methods.

Although brief, the discussion provides a nice summary of the key findings and implications of the study. The appendices offer further details in explaining certain concepts and findings while also instructing on how to implement the techniques discussed throughout the manuscript.

Pataky TC. One-dimensional statistical parametric mapping in Python. Comput Methods Biomech Biomed Engin. 2012;15(3):295-301.

Experimental design

The manuscript clearly defines the research questions in the introduction, and these questions are thoroughly answered without going outside of the scope of the study. The experimental design is also appropriate for answering these questions. The simulated dataset provides a theoretical framework for highlighting the link between 0D and extended ROI 1D methods, while the publically available datasets allow the authors to present and discuss different practical ways for implementing 1D ROI analyses.

Validity of the findings

The findings detailed throughout the manuscript are well supported by the data, and presented in an informative and appropriate matter. As noted earlier in the “Basic Reporting” comments, the findings are well backed by data and appropriate explanations / use of figures. All of the claims made by the authors through the manuscript are supported by the experimental findings.

Additional comments

The authors should be commended on the production of this manuscript; as it is a concise, easy to understand description of what could be considered a complex analytical topic. I believe those interested in alternative statistical methods will appreciate this manuscript, while it also has the potential to introduce others to a novel technique for analysing nD datasets. This is an important paper for increasing understanding around 1D methods to promote their use across scientific research. A potential misunderstanding of 1D methods may be that they are the result of repetitive 0D tests across a field of interest. This paper does an excellent job of explaining how this is not the case, but rather that augmenting the ROI for 1D analysis is an extension of 0D methods.

I would recommend minimal changes to this manuscript, with the exception of potentially including some extra details / references specific to the SPM method in the introduction (as noted in the “Basic Reporting” comments).

Reviewer 2 ·

Basic reporting

In this manuscript, the authors describe the increase in statistical power provided by the restriction of the search volume when trying to detect significant effects in biomechanical measurements using statistical parametric mapping. While all the results presented here using the random field theory are well known and well established in the neuroimaging field (from which they are borrowed), they have only been recently introduced in the biomechanical literature and are therefore of interest to this community. The paper is short, well written with figures of good quality, and the appendix gives nice step-by-step instructions on how to perform similar analyses with their open source toolbox.

Coming from the neuroimaging field, this manuscript was hard to read at first, due to the choice of wording, despite the authors trying to clarify their nomenclature in the introduction of the paper. What the authors mean by “ROI analysis” is what others call “Small Volume Correction” (SVC; applying nD method on $S_0 \subset S$, as described in [1]). Instead, a ROI analysis usually refers to the specification of a region of interest ($S_{VOI} \subset S$), summarising the signal within it (typically a simple average) and performing a 0D method.

I would also argue that the general approach is nD, with ROI and 0D being special cases (when search space S reduces (ie, ROI approach), up to a single measurement where the nD approach boils down to computing the uncorrected p-value for that measurement (ie, 0D approach)). This is in essence the unified approach presented in [1].

The authors should devote a significant section of the manuscript on the issues relating to the definition of the ROI: while the gain in statistical power is a natural consequence of the reduction in search volume, the inference can be biased if the definition of the search volume (ROI or 0D) is not orthogonal to the effect of interest. Again, the dangers of circular analyses and double dipping are well-documented (see eg [2] and [3]) but they are worth reemphasizing here (they are somehow acknowledged in the discussion). The ROI can be defined using orthogonal data (as in dataset D with a pilot study) or orthogonal contrasts, but not using a local extremum from the data themselves (as in datasets A and B). This might also be an issue in dataset C if the ROI is centred at the extremum of the average of the side-shuffle and V-cut manoeuvers: the differential effect will only be orthogonal to the average effect if there are the same numbers of trials in both conditions (see the supplementary material in [3] about orthogonal contrasts).

An example of danger of circularity could actually be formed using datasets A/B: looking at the data in figure 1, one could define a ROI at or around the local maximum at time=15%, that would be spuriously declared as significant according to the threshold displayed in figure 4. In contrast, the 1D approach rightly fails to reject the null hypothesis in this region when searching for an effect in the entire space.

While the code for the analyses is available as a toolbox in two computer languages, there are no mention of the availability of the experimental datasets (C and D).

[1] K.J. Worsley et al, A unified statistical approach for determining significant signals in images of cerebral activation. Human Brain Mapping, 1996.
[2] J.M. Kilner, Bias in a common EEG and MEG statistical analysis and how to avoid it. Clinical Neurophysiology, 2013.
[3] N. Kriegeskorte et al, Circular analysis in systems neuroscience: the dangers of double dipping. Nature Neuroscience, 2009.
[4] K.J. Friston et al, A critique of functional localisers. Neuroimage, 2006.
[5] R. Saxe et al, Divide and conquer: a defense of functional localizers. Neuroimage, 2006.

Experimental design

No Comments.

Validity of the findings

No Comments.

Additional comments

Minor issues:
* l.24: replace \alpha with type I error rate or significance level in the introduction
* For all datasets, mention the number of time points and degrees of freedom.
* Figure 3: put the legend in both plots (or in the first one instead of the second one only).
* Some of the references from l.174 concern the differences between nD and 0D, and not ROI. The references [4] and [5] might also be relevant.
* References: {MarsBar}, {SPM99}, {PCA}, {SPM}, {Python}, {fMRI}
* Appendix B.1, MATLAB example: it should probably be spm1d.stats.ttest2(YB, YA, …) and t.inference(0.05)

---

## Round 0.2 · Minor Revisions

The reviewer has checked the Appendix and made some helpful comments on that and some other parts of the paper. These are relatively minor and thus easy to fix. We will accept the paper once they are taken into account, so please make amendments when you are ready.

Reviewer 2 ·

Basic reporting

No Comments

Experimental design

No Comments

Validity of the findings

No Comments

Additional comments

Thanks to the authors for quick and clear responses to the comments made in the first round of review.

The new appendix A is a very thorough attempt at clarifying the terminology used in the manuscript and its relation to the one used in the neuroimaging literature. I just hope it won't have the opposite effect of confusing readers by its apparent complexity! I agree with the authors that ROI and VOI are often used interchangeably - the reason behind this is that one can consider a VOI to be a special case of a ROI in 3D. I would therefore simplify Table A.2 to have ROI and VOI equivalent (and the same for ROI and VOI analyses) and oppose them to a SVC analysis (and ignore the inconsistency of the SPM software in its documentation).

The new paragraph in the introduction (line 68) somehow implies that the SPM approach has mainly been used in 3D in neuroimaging but this would be ignoring electrophysiology (EEG, MEG), see e.g.:
[1] Kilner et al, Applications of random field theory to electrophysiology. Neuroscience Letters, 2004.
[2] Kiebel and Friston, Statistical parametric mapping for event-related potentials: I. Generic considerations. NeuroImage, 2004.
[3] Kilner and Friston, Topological inference for EEG and MEG. The Annals of Applied Statistics, 2010.

I still think the dangers of circular analyses are crucial here and have to be highlighted. The authors have added an extra paragraph in the discussion so that might be enough for the moment. The following might be an interesting reference to the authors:
[4] Brooks et al, Data-driven region-of-interest selection without inflating Type I error rate. Psychophysiology, 2016.

At last, sorry for the unclear comment about references ({MarsBar}, {fMRI}, ...), I was simply pointing out that some letters had to be upper case - it might be an issue with the PeerJ formatting though.

---

## Round 0.3 · accepted · Accept

Congratulations on your excellent paper's acceptance! Thanks for your diligent revisions and your patience with the review process.

---

## Author Rebuttal · Round 0.3

**PeerJ #2016:06:11626:0 (Review #2)**

"Region-of-interest analyses of one-dimensional biomechanical trajectories: bridging 0D and 1D theory, augmenting statistical power"

We thank the Editors and Reviewer #2 once again for your time. Please find that we have responded to all comments below using blue text and that we have highlighted changes to the manuscript in yellow.

Thank you,

Todd Pataky, Mark Robinson, and Jos Vanrenterghem
* * *
## Comments for the author

The new appendix A is a very thorough attempt at clarifying the terminology used in the manuscript and its relation to the one used in the neuroimaging literature. I just hope it won't have the opposite effect of confusing readers by its apparent complexity! I agree with the authors that ROI and VOI are often used interchangeably - the reason behind this is that one can consider a VOI to be a special case of a ROI in 3D. I would therefore simplify Table A.2 to have ROI and VOI equivalent (and the same for ROI and VOI analyses) and oppose them to a SVC analysis (and ignore the inconsistency of the SPM software in its documentation).
Response:  We agree, all suggested changes have been made to Table A.2 and its description.  Please find that we have also removed the final paragraph of the Introduction (main manuscript) because we agree that this may unnecessarily complicate the issue.

The new paragraph in the introduction (line 68) somehow implies that the SPM approach has mainly been used in 3D in neuroimaging but this would be ignoring electrophysiology (EEG, MEG), see e.g.:
[1] Kilner et al, Applications of random field theory to electrophysiology. Neuroscience Letters, 2004.
[2] Kiebel and Friston, Statistical parametric mapping for event-related potentials: I. Generic considerations. NeuroImage, 2004.
[3] Kilner and Friston, Topological inference for EEG and MEG. The Annals of Applied Statistics, 2010.
Response:  We agree.  Please find that we have added citations to the Kilner (2004) and Kiebel (2004) papers, and that we have revised our phrasing to emphasize SPM's scope beyond neuroimaging.

I still think the dangers of circular analyses are crucial here and have to be highlighted. The authors have added an extra paragraph in the discussion so that might be enough for the moment. The following might be an interesting reference to the authors:

[4] Brooks et al, Data-driven region-of-interest selection without inflating Type I error rate. Psychophysiology, 2016.

Response: Thank you for re-emphasizing this point, we agree that it is a very important issue. Please find that we have added a citation to Brooks (2016) and have also added another paragraph regarding circularity to the Discussion.

At last, sorry for the unclear comment about references ({MarsBar}, {fMRI}, ...), I was simply pointing out that some letters had to be upper case - it might be an issue with the PeerJ formatting though.

Response: Thank you for clarifying, we see now what you mean. This indeed appears to have been caused by PeerJ's CLS formatting. We have overridden the CLS formatting to fix all terms, and we'll re-check all during preprint checks.